# Air pollution measurement errors: Is your data fit for purpose?

Sebastian Diez[1], Stuart E. Lacy[1], Thomas J. Bannan[2], Michael Flynn[2], Tom Gardiner[3], David Harrison[4], Nicholas Marsden[2], Nick Martin[3], Katie Read[1,5], Pete M. Edwards[1]

[1]Wolfson Atmospheric Chemistry Laboratories, University of York, York, YO10 5DD, UK

[2]Department of Earth and Environmental Science, Centre for Atmospheric Science, School of Natural Sciences, The University of Manchester, Manchester, M13 9PL, UK

[3]National Physical Laboratory, Teddington TW11 0LW, UK

[4]Bureau Veritas UK, London, E1 8HG, UK

[5]National Centre for Atmospheric Science, University of York, York, YO10 5DD, UK

*Correspondence:*

Sebastian Diez (sebastian.diez@york.ac.uk); Pete Edwards (pete.edwards@york.ac.uk)

**Abstract.** When making measurements of air quality, having a reliable estimate of the measurement uncertainty is key to assessing the information content that an instrument is capable of providing, and thus its usefulness in a particular application. This is especially important given the widespread emergence of Low Cost Sensors (LCS) to measure air quality. To do this, end users need to clearly identify the data requirements a priori and design quantifiable success criteria by which to judge the data. All measurements suffer from errors, with the degree to which these impact the accuracy of the final data often determined by our ability to identify and correct for them. The advent of LCS has provided a challenge in that many error sources show high spatial and temporal variability, making laboratory derived corrections difficult. Characterising LCS performance thus currently depends primarily on colocation studies with reference instruments, which are very expensive and do not offer a definitive solution but rather a glimpse of LCS performance in specific conditions over a limited period of time. Despite the limitations, colocation studies do provide useful information on measurement device error structure, but the results are non-trivial to interpret and often difficult to extrapolate to future device performance. A problem that obscures much of the information content of these colocation performance assessments is the exacerbated use of global performance metrics ($R^2$, RMSE, MAE, etc.). Colocation studies are complex and time-consuming, and it is easy to fall into the temptation to only use these metrics when trying to define the most appropriate sensor technology to subsequently use. But the use of these metrics can be limited, and even misleading, restricting our understanding of the error structure and therefore the measurements' information content. In this work, the nature of common air pollution measurement errors is investigated, and the implications these have on traditional metrics and other empirical, potentially more insightful, approaches to assess measurement performance. With this insight we demonstrate the impact these errors can have on measurements, using a selection of LCS deployed alongside reference measurements as part of the QUANT project, and discuss the implications this has on device end-use.

34

## 1. Introduction

The measurement of air pollutants is central to our ability to both devise and assess the effectiveness of policies to improve air quality and reduce human exposure (Molina & Molina, 2004). The emergence of low-cost sensor (LCS) based technologies means a growing number of measurement devices are now available for this purpose (Morawska et al., 2018), ranging from small low-cost devices that can be carried on an individual's person all the way through to large, expensive reference and research-grade instrumentation. A key question that needs to be asked when choosing a particular measurement technology is whether the data provided is fit for purpose (Andrewes et al., 2021; Lewis & Edwards, 2016). In order to answer this, the user must first clearly define the question that is to be asked of the data, and thus the information required. For example, a measurement to characterize "rush hour" concentrations, or to determine if the concentration of a pollutant exceeded an 8 h average legal threshold value at a particular location would demand a very different set of data requirements than a measurement to determine if a change in policy had modified the average pollutant concentration trend in a neighbourhood. Would the $R^2$ or RMSE or any other global single-value metric be enough to decide between the different device's options? Considerations such as the origin of the performance data, type of experiment (laboratory or colocation) (Jiao et al., 2016), the test location (Feenstra et al., 2019) and period (i.e. duration, season, etc.), the LCS and reference measurement method (Giordano et al., 2021), measurement time resolution and ability to capture spatial variability (Feinberg et al., 2019) would be important factors to consider for such examples. The measurement uncertainty is also of critical consideration, as this ultimately determines the information content of the data, and hence how it can be used (Tian et al., 2016).

All measurements have an associated uncertainty, and even in highly controlled laboratory assessments, the true value is not known, with any measurement error defined relative to our best estimate of the range of possible true values. However, quantifying and representing error and uncertainty is a challenge for a wide range of analytical fields, and often what these concepts represent is not the same to all practitioners. This results in a spectrum of definitions that take into account the way truth, error, and uncertainty are conceived (Grégis, 2019; Kirkham et al., 2018; Mari et al., 2021). For atmospheric measurements assessing uncertainty is complex and non-trivial. Firstly, given the "true" value can never be known, an agreed reference is needed. Secondly, the constantly changing atmospheric composition means that repeat measurements cannot be made and the traditional methods for determining the random uncertainty are not applicable. And finally, a major challenge arises from the multiple sources of error both internal and external to the sensor that can affect a measurement. Signal responses from a non-target chemical or physical parameter or electromagnetic interference are examples of an almost limitless number of potential sources of measurement error. In this work, we will follow the definitions given by the International Vocabulary of Metrology (JCGM, 2012) for measurement error ("measured quantity value minus a reference quantity value") and for measurement uncertainty ("non-negative parameter characterising the dispersion of the quantity values being attributed to a measurand, based on the information used"). Also, when the term "uncertainty" is used here, it is referring to "diagnosis uncertainty", in contrast with "prognosis uncertainty" (see Sayer et al., 2020 for more details).

The covariance of many of the physical and chemical parameters of the atmosphere, makes accurately identifying
particular sources of measurement interference or error very difficult in the real world. Unfortunately, specific
laboratory experiments for the characterization of errors is complex and very expensive, resulting in many sources
of error being essentially unknown for many measurement devices. The use of imperfect error correction
algorithms that are not available to the end-user (e.g. in many LCS devices) makes error identification and
quantification even more complex. For this reason, colocation experiments in relevant environments are often the
best option to assess the applicability of a given measurement method for its intended purpose.
The mentioned difficulties in defining and quantifying uncertainty across the full range of end-use applications of
a measurement device, means that often the quoted measurement uncertainty is not applicable, or in some cases
not provided or provided in an ambiguous manner. This makes assessing the applicability of a measurement device
to a particular task difficult for users. In this work, we investigate the nature of common air pollution measurement
errors, and the implications these have on traditional goodness-of-fit metrics and other, potentially more insightful
approaches to assess measurement uncertainty. We then use this insight to demonstrate the impact these errors
can have on measurements, using a selection of LCS deployed alongside reference measurements as part of the
UK Clean Air program funded QUANT (Quantification of Utility of Atmospheric Network Technologies) project,
a 2-year colocation study of 26 commercial LCS devices (56 gases measurements and 56 PM measurements) at
multiple urban, background and roadside locations in the UK. After analysing some of the real-life uncertainty
characteristics we discuss the implications this has on data use.
**2. Error characterization**
When characterising measurement error, in the absence of evidence to the contrary a linear additive model is often
assumed. Once the analytical form of the model is defined, its parameters aim to capture the error characteristics,
and in the case of linear models (Eq. (1)), these are typically separated into three types (Tian et al., 2016): (i)
proportional bias or scale error ($b_1$), (ii) constant bias or displacement error ($b_0$) and (iii) random error ($\varepsilon$) (Tian et
al., 2016). Any measurement ($y_i$, e.g from the LCS) can therefore be thought of as a combination of the reference
value ($x_i$) and the three error types, such that:
$$y_i = b_1 x_i + b_0 + \varepsilon \tag{1}$$
As the simplest approximation, this linear relationship for the error characteristics is often used to correct for
observed deviations between measurements and the agreed reference. It is worth to note, however, that this
equation assumes time-independent error contributions and that the three error components are not correlated,
which is often not the case on both counts (e.g. responses to non-target compounds). The parameter values
determined for Eq. (1) are also generally only applicable for individual instruments, potentially in specific
environments, unless the transferability of these parameters between devices has been explicitly demonstrated.
Figure 1 shows examples of how pure constant bias (a-panels), pure proportional bias (b-panels), and pure random
noise (c-panels) would look like in time-series, regression, Bland-Altman (B-A) (Altman & Bland, 1983) and
Relative Expanded Uncertainty (REU, as defined by the GDE (2010)) plots. In each of these ideal cases, the error
plots enable the practitioner to view the error characteristics in slightly different ways, allowing the impacts of the
observed measurement uncertainty to be placed into the context of the data requirements. In this work, we will

refer to them as "error types" (in contrast to "error sources"), which is the way they are distilled by the linear error model.

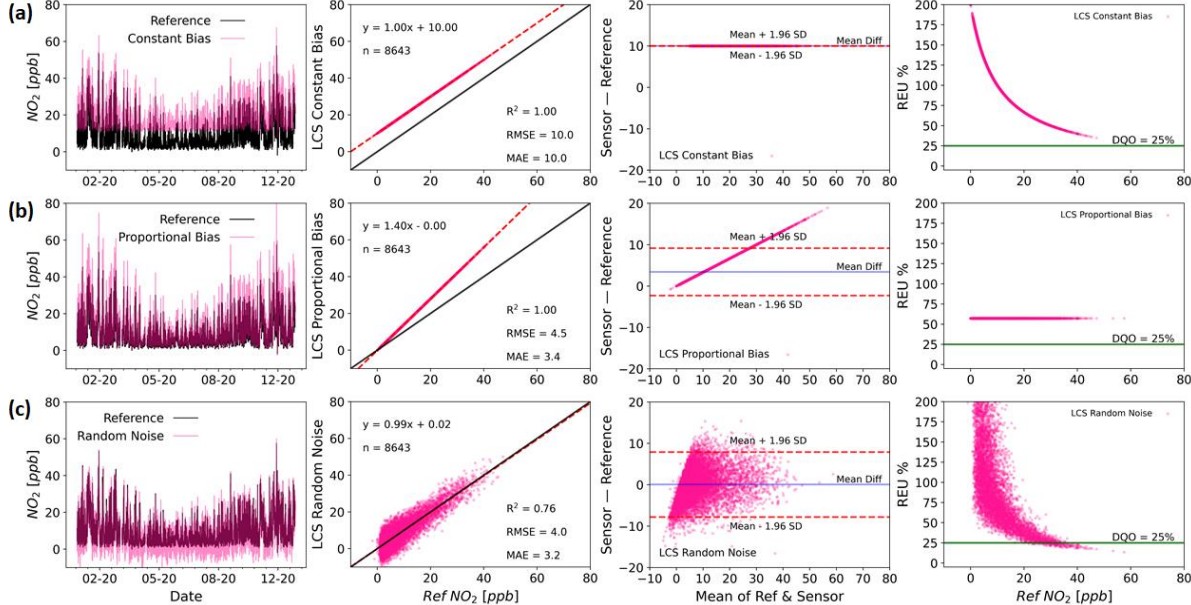

**Figure 1. Time series (left panels), regression (middle-left panels), B-A Bland-Altman (middle-right panels) and REU (right panels, DQO for NO₂ = 25%) plots for arbitrary examples of pure constant bias (Slope = 1, Intercept = 1, SDε = 0; a-panels), pure proportional bias (Slope = 1.4, Intercept = 0, SDε = 0; b-panels) and pure random noise (Slope = 1, Intercept = 0, SDε = 4; c-panels) simulated errors.**

## 2.1 Performance indices, error structure and uncertainty

A major challenge faced by end-users of measurement devices characterised using colocation studies is the non-trivial question of how the comparisons themselves are performed and how the data are ~~is~~ communicated. Often single value performance metrics, such as the coefficient of determination ($R^2$) or root mean squared error (RMSE), are calculated between the assessed method (e.g. LCS) and an agreed reference, and the user is expected to infer an expected device performance or uncertainty for a measurement in their application (Duvall et al., 2016; Malings et al., 2019). When evaluating multiple sensors during a colocation experiment, single metrics can be a useful way to globally compare instruments/sensors. However, these metrics do little to communicate the nature of the measurement errors and the impacts these will have in any end use application, in part because they reduce the error down to a single value (Tian et al., 2016). Even more if a specific concentration range is of paramount interest to the end-user, these metrics are not capable of characterising the weight of noise and/or the bias effect. The $R^2$ shows globally the data set linearity and gives an idea of the measurement noise. However, it is unable to distinguish whether a specific range of concentrations is more or less linear (or more or less noisy) than another. Similarly, the RMSE is also a very useful metric and perhaps more complete than $R^2$, as it considers both noise and bias (although they need to be explicitly decomposed from RMSE). Nevertheless, the RMSE is an average measure (of noise and bias) over the entire dataset under analysis. Using combinations of simple metrics increases the information communicated, but does not necessarily make it easy to assess how the errors will likely impact

a particular measurement application. Visualising the absolute and relative measurement errors across the
concentration range (unreachable by global metrics) enables end users to view the errors, and any features (non-
linearities, step changes, etc.) that would impact the measurement but that global metrics (and in some cases time-
series and/or regression plots) are incapable of showing.
Unfortunately, the widespread use of a small number of metrics as the sole method to assess measurement
uncertainty, without a thorough consideration of the nature of the measurement errors, means measurement
devices are often chosen that are unable to provide data that is fit for purpose. In addition, unconscious about
potential flaws, users (e.g. researchers) could communicate findings or guide decision making based on results
that may not justify the conclusions drawn from the data. Figure 2 shows three simulated measurements compared
with the true values. Despite the measurements having identical $R^2$ and RMSE values, the time series and
regression plots show that the error characteristics are significantly different, and would impact how the data from
such a device could viably be used.

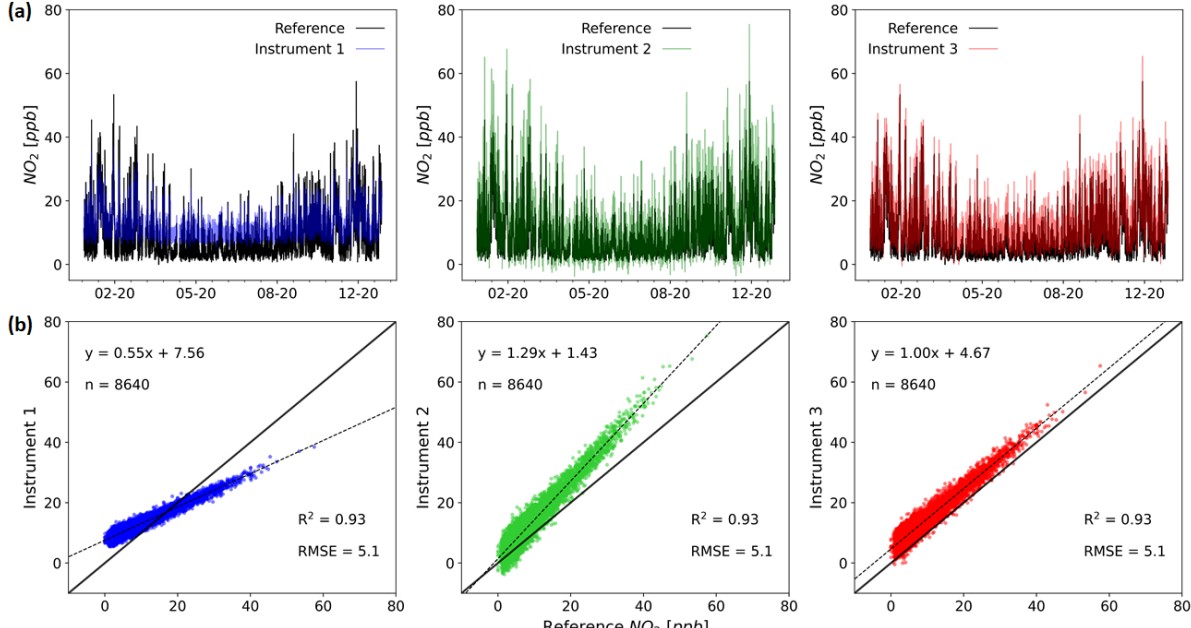

**Figure 2. Time series (a-panels) and regression plots (b-panels) for three hypothetical instruments and a reference (1**
**year of data). The most used metrics for evaluating the performance of LCS ($R^2$ and RMSE) are identical for the**
**systems shown, even when the errors have very different characteristics (time res 1 h).**
There are multiple performance metrics that can be used for the assessment of measurement errors and uncertainty.
Tian et al (2016) present an excellent summary of some of the major pitfalls of performance metrics and promote
an approach of error modelling as a more reliable method of uncertainty quantification. These modelling
approaches, however, rely on the assumption of statistical stationarity, whereby the statistical properties of the
error are constant in the temporal and spatial domains. The presence of unknown or poorly characterised sources
of error, for example, due to interferences from other atmospheric constituents or drifts in sensor behaviour, makes
this assumption difficult to satisfy, especially when the dependencies of these errors show high spatial and
temporal variability. Thus, if field colocation studies are the primary method for performance assessment, as is
the case for LCS, only through a detailed assessment of the measurement errors across a wide range of conditions
and timescales can the uncertainty of the measurement be realistically estimated.
**2.2 Dealing with errors: established techniques vs Low-Cost Sensors**
Different approaches are available to the user to minimise the impact of errors, generally by making corrections
to the sensor data. For example, in the case of many atmospheric gas analysers, if the error is dominated by a
proportional bias, a multi-point calibration can be performed using standard additions of the target gas.
Displacement errors can be quantified, and then corrected for, by sampling a gas stream that contains zero target
gas. Random errors can be reduced by applying a smoothing filter (e.g moving average filter, time-averaging the
data, etc.), at the cost of losing some information (Brown et al., 2008). These approaches work well for simple
error sources that, ideally, do not change significantly over timescales from days to months. Unfortunately, more
complex error sources can manifest in such a way that they contribute across all three error types, and also vary
temporally and spatially. For example, an interference from another gas-phase compound could in part manifest
itself as a displacement error, based on the instrument response to its background value, and in part as a
proportional bias if its concentration correlates with the target compounds, with any short-term deviations from
perfect correlation contributing to the random error component. In this case, time-averaging combined with
periodic calibrations and zeros would not necessarily minimise the error, and the user would need to employ
different tactics. One option would be to independently measure the interferent concentration, albeit with
associated uncertainty, and then use this to derive a correction. This is feasible if a simple and cost-effective
method exists for quantifying the interferent and its influence on the result is understood, but can make it very
difficult to separate out error sources, and can become increasingly complex if this measurement also suffers from
other interferences.
For many measurement devices, in particular for LCS based instruments, a major challenge is that the sources and
nature of all the errors are unknown or difficult to quantify across all possible end-use applications, meaning
estimates of measurement uncertainty are difficult. In the case of most established research and reference-grade
measurement techniques, comprehensive laboratory and field experiments have been used to explore the nature
of the measurement errors (Gerboles et al., 2003; Zucco et al., 2003). Calibrations have then been developed,
where traceable standards are sampled and measurement bias, both constant and proportional, can be corrected
for. Interferences from variables such as temperature, humidity, or other gases, have also been identified and then
either a solution engineered to minimise their effect or robust data corrections derived. Unfortunately, these
approaches have been shown not to perform well in the assessment of LCS measurement errors, due to the
presence of multiple, potentially unknown, sensor interferences from other atmospheric constituents (Thompson
& Ellison, 2005). These significant sensitivities to constituents such as water vapour and other gases mean
laboratory-based calibrations of LCS become exceedingly complex, and expensive, as they attempt to simulate
the true atmospheric complexity, often resulting in observed errors being very different to real-world sampling
(Rai et al., 2017; Williams, 2020). This has resulted in colocation calibration becoming the accepted method for
characterising LCS measurement uncertainties (De Vito et al., 2020; Masson et al., 2015; Mead et al., 2013;
Popoola et al., 2016; Sun et al., 2017), where sensor devices are run alongside traditional reference measurement
systems for a period of time, and statistical corrections derived to minimise the error between the two. As the true
value of a pollutant concentration cannot be known, this colocation approach assumes all the error is in the low-

cost measurement. Although this assumption may often be approximately valid (i.e. reference error variance << LCS error variance), no measurement is absent of uncertainty and this can be transferred from one measurement to another, obscuring attempts to identify its sources and characteristics. A further consideration when the fast time-response aspect of LCS data is important, is that reference measurement uncertainties are generally characterised at significantly lower reported measurement frequencies (typically 1 hr). This means that a high time-resolution (e.g. 1 min) reference uncertainty must be characterised in order to accurately estimate the LCS uncertainty (requiring specific experiments and additional costs). If a lower time-resolution reference data set is used as a proxy, then the natural variability timescales of the target compound should be known and any impact of this on the reported uncertainty caveated.

Another challenge with this approach is that, unlike targeted laboratory studies, real-world colocation studies at a single location, and for a limited time period, are not able to expose the measurement devices to the full range of potential sampling conditions. As many error sources are variable, both spatially and temporally, using data generated under a limited set of conditions to predict the uncertainty on future measurements is risky. Deploying a statistical model makes the tacit assumption that all factors affecting the target variable are captured by the model (and the data set used to build the model). This is very often an unrealistic demand, and in the complex multifaceted system that is atmospheric chemistry, this is extremely unlikely to be tenable, resulting in a clear potential for overfitting to the training dataset. Ultimately, however, these colocation comparisons with instruments with a well-quantified uncertainty need to be able to communicate a usable estimate of the information content of the data to end-users, so that devices can be chosen that are fit for a particular measurement purpose.

### 3. Methods

In this work, we explore measurement errors, and their impacts, using the most common single value metrics: the Coefficient of Determination or $R^2$, the Root Mean Squared Error or RMSE and the Mean Absolute Error or MAE (see the equation definitions in Cordero et al., 2018). To visualise the error distribution across a dataset we have also employed two additional widely used approaches: the Bland-Altman plots (B-A) and Relative Expanded Uncertainty (REU).

The performance metrics provide a single value irrespective of the size of the dataset, and might appear convenient for users when comparing across devices or datasets, but can encourage over-reliance on the metric, often at the expense of looking at the data in more detail or bringing an awareness of the likely physical processes driving the error sources. On the other hand, the use of visualisations such as B-A and REU is complementary to the aforementioned metrics, with the added value that the user is now more aware of how the data looks like in an absolute and/or relative error space, allowing them to distinguish some characteristics of interest. These visualizations are indeed more laborious and the interpretation can be challenging for non-experts, but they provide additional insights into the nature of the errors, not attainable by one or more combined performance metrics: while B-A plots shows the noise (dispersion of the data) and the bias effect (tendency of the data) in an absolute scale, the REU can be explicitly decomposed in the noise and bias components (see Yatkin et al., 2022).

In order to understand how the different tools used here show different characteristics of the error structure, some errors commonly found in LCS are examined through simulation studies. Subsequently, two real world case

studies are presented: (i) LCS duplicates for $NO_2$ and $PM_{2.5}$ belonging to the QUANT project located in two sites
-the Manchester Natural Environment Research Council (NERC) measurement Supersite, and the York Fishergate
Automatic Urban and Rural Network (AURN) roadside site- and (ii) a set of duplicate reference instruments (only
at Manchester Supersite). Table S1 shows the research grade instrumentation used for this study.

### 3.1 Visualisation tools

An ideal performance metric should be able to deliver not only a performance index but also an idea of the
uncertainty distribution (Chai & Draxler, 2014). This is difficult to deliver through a simple numerical value, and
easy to interpret visualisations of the data are often much more useful for conveying multiple aspects of data
performance. Figure 2 shows the two most common data visualisation tools, the time-series plot and the regression
plot. In the time series plot the instrument under analysis and the agreed reference are plotted together as a function
of time. This allows a user to visually assess tendencies of over or under prediction, differences in the base line
or other issues, but can be readily over interpreted and does not allow for easy quantification of the observed
errors. In the regression plot the data from the instrument under analysis is plotted against the agreed reference
data. This allows for the correlation between the two methods to be more readily interpreted, in particular any
deviations from linearity, but gives little detail on the nature of the errors themselves.
In contrast to the regression plot -where the measured values from the two measurements (e.g. LCS vs Ref) are
plotted against each other- the Bland-Altman plots essentially display the difference between measurements
(abscissa) as a function of the average measurement (ordinate), enabling more information on the nature of the
error to be communicated. This direct visualisation of the absolute error acknowledges that the true value is
unknown and that both measurements have errors. The B-A plot enables the easy identification of any systematic
bias between the measurements or possible outliers, and is the reason B-A plots are extensively used in analytical
chemistry and biomedicine to evaluate agreement between measurement methods (Doğan, 2018). The mean
difference between the measurements, (represented by the blue line in the figures), is the estimated bias between
the two observations. The spread of error values around this average line indicates if the error shows purely
random fluctuations around this mean, or if it has structure across the observed concentration range.
In the case where all the error is assumed to be in one of the measurements, e.g. comparing a LCS to a reference
grade measurement, there is an argument that the B-A abscissa could be the agreed reference value instead of the
average of two measurements. However, in this work we use the average of the two values as per the traditional
B-A analysis. To illustrate the B-A interpretation, from the error model (Eq. (1)) we can derive the following
expression:
$$y_i - x_i = x_i\,(b_1 - 1) + b_0 + \varepsilon \tag{2}$$
From Eq. (2) it can be seen that if $b_1 \neq 1$ or if the error term ($\varepsilon$) variance is non-constant (e.g. heteroscedasticity)
the difference will not be normally distributed. The B-A plot (with $x_i$ as the reference instrument results) allows a
quick visual assessment of the error distribution without the need to calculate the model parameters. In the case
the differences are normally distributed, the so-called "agreement interval" (usually defined as $\pm\,2\sigma$ around the
mean) will hold 95% of the data points. Even though the estimated limits of agreement will be biassed if the
differences are not normally distributed, it can still be a valuable indicator of agreement between the two
measurements.
If the ultimate goal of studying measurement errors is to diagnose the measurement uncertainty in a particular
target measurement range, then visualising the uncertainty in pollutant concentration space can be very
informative. The REU provides a relative measure of the uncertainty interval about the measurement within which
the true value can be confidently asserted to lie. The abscissa in an REU plot represents the agreed reference
pollutant concentration, whose error is taken into account, something not considered by the other metrics or
visualisations discussed. The REU is regularly used to assess measurement compliance with the Data Quality
Objective (DQO) of the European Air Quality Directive 2008/50/EC, and is mandatory for the demonstration of
equivalence of methods other than the EU reference methods. For LCS the REU is widely used as a performance
indicator (Bagkis et al., 2021; Bigi et al., 2018; Castell et al., 2017; Cordero et al., 2018; Spinelle et al., 2015).
However, the evaluation of this metric is perceived as arduous and cumbersome and it is not included in the
majority of sensor studies (Karagulian et al., 2019). There is now a new published European Technical
Specification (TS) for evaluating the LCS performance for gaseous pollutants (CEN/TS 17660-1:2021). It
categorises the devices in 3 classes according to the DQO (Class 1 for "indicative measurements", Class 2 for
"objective estimations", and Class 3 for non-regulatory purposes, e.g. research, education, citizen science, etc.).
In the following sections, we use these established methods for assessing measurement uncertainty, alongside
simple time series and regression plots, to explore different error sources and their implications for air pollution
measurements.

**4. Case studies**

**4.1 Simulated instruments**

In order to investigate the impact of different origins of measurement error on measurement performance, a set of
simulated datasets have been created. These data are derived using real-world reference data as the true values,
with the subsequent addition of errors of different origins to generate the simulated measurement data. Error
origins were chosen for which examples have been described in the LCS literature. Performance metrics along
with visualisation methods are then used to assess measurement performance.
As the complexity of the error increases, the impact of the assumption of statistical stationarity can become more
difficult to satisfy, with the magnitude of the errors becoming less uniform across the observed concentration, and
hence spatial, or time domains. Figure 3 shows examples of modelled sources of errors on $NO_2$ measurements:
temperature interference (correction model taken from (Popoola et al., 2016), a-panels), a non-target gas (ozone)
interference (correction model taken from (Peters et al., 2021), b-panels) and thermal electrical noise (white noise,
c-panels).

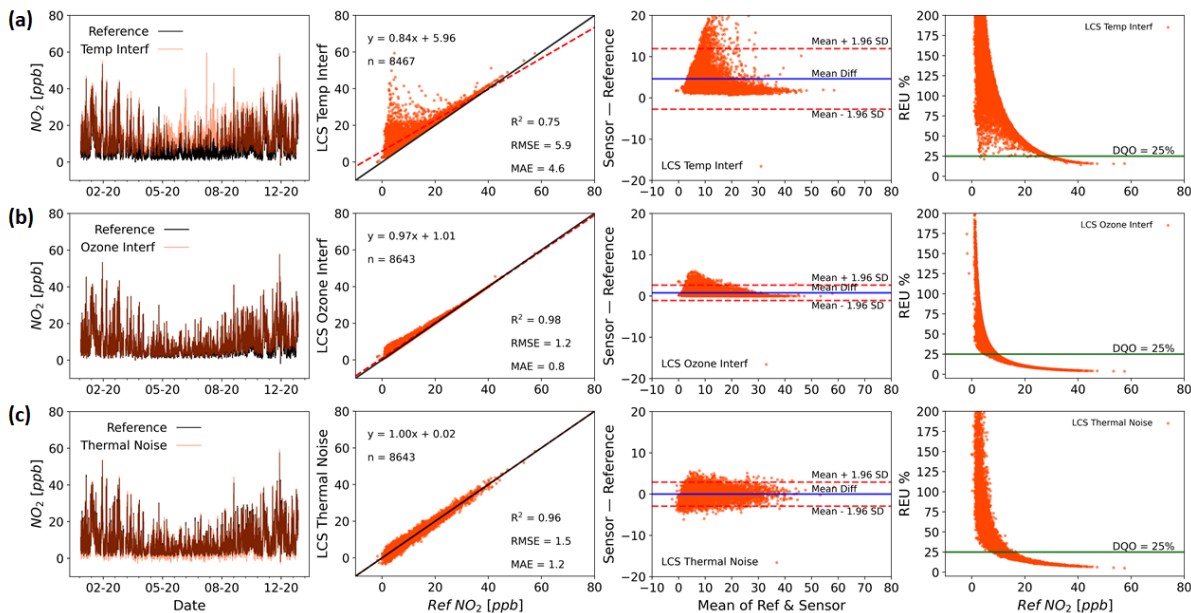

**Figure 3. Time series (left panels), regression plots (middle-left panels, including R², RMSE & MAE), Bland-Altman plots (middle-right panels) and REU (right panels, DQO for NO₂ = 25%) for temperature (a-panels), ozone (b-panels) and thermal electrical noise (c-panels) modelled interferences on NO₂ measurements (time res 1 h).**

The above simulations show examples of how individual sources of error can impact measurement performance. Figure S1 shows some more examples, this time for different drift effects (baseline drift, temperature interference drift and instrument sensitivity drift). This set of error origins is not exhaustive, with countless others potentially impacting the measurement, such as those coming from (i) hardware (sensor-production variability, sampling, thermal effects due to materials expansion, drift due to ageing, RTC lag, Analog-to-Digital conversion, electromagnetic interference, etc.), (ii) software (signal sampling frequency, signal-to-concentration conversion, concept drift, etc.), (iii) sensor technology/measurement method (selectivity, sensitivity, environmental interferences, etc.) and (iv) local effects (spatio-temporal variation of concentrations, turbulence, sampling issues etc.).

Each error source impacts the uncertainty of the measurement, which in turn impacts its ability to provide useful information for a particular task. For example, the form of the temperature interference shown in Fig. 3 (a-panels) results in the largest errors being seen at the lower NO₂ values. This is because NO₂ concentrations are generally lowest during the day, due to photolytic loss when temperatures are highest. Thus this device would be better suited to an end-user intending to assess daily peak NO₂ concentration compared with the daytime hourly exposure values, providing the environment the device was deployed in showed a similar relationship between temperature and true NO₂ as that used here. The O₃ interference shown in Fig. 3 (b-panels) is similar, due again to a general anti-correlation observed between ambient O₃ and NO₂ concentrations. This type of interference can often be interpreted incorrectly as a proportional bias, and a slope correction applied to the data. However, this type of correction will ultimately fail as O₃ concentrations are dependent on a range of factors, such as hydrocarbon concentrations and solar radiation, and as these change the O₃ concentration relative to the NO₂ concentration will change. To further complicate matters, multiple error sources can act simultaneously, meaning that the majority

of measurements will contain multiple sources of error. Figure 4 shows a simple linear combination of the
modelled errors shown in Fig 3, and the impact this has on the performance metrics.

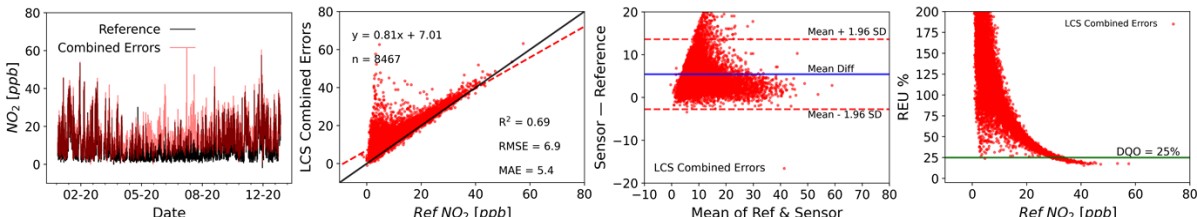


**Figure 4. Time series (left panel), regression plot (middle-left panel, including $R^2$, RMSE & MAE), Bland-Altman**
**plot (middle-right panel) and REU (right panel, DQO for $NO_2$ = 25%) for a linear combination of temperature, ozone**
**and thermal electrical noise modelled interferences (time res 1 h).**
As the simulations show, the nature of the errors determine the observed effect on the measurement performance.
In an ideal situation, like those shown in figures 3 and 4, the error sources would be well characterised, allowing
the error to be modelled and approaches such as calibrations (for bias) and smoothing (for random errors)
employed to minimise the total uncertainty. Unfortunately, in scenarios where sources of error and their
characteristics are not known, modelling the error becomes more difficult and a more empirical approach to
assessing the measurement performance and uncertainty may be required. The growing use of LCS represents a
particular challenge in this regard. The susceptibility of LCS to multiple, often unknown or poorly characterised,
error sources means that in order to determine if a particular LCS is able to provide data with the required level
of uncertainty for a given task, a relevant uncertainty assessment is required. The following section explores the
uncertainty characteristics of several LCS, with unknown error sources, deployed alongside reference
instrumentation in UK urban environments as part of the QUANT study.
**4.2 Real-world instruments**
The difficulty in generating representative laboratory error characterisation data means for many measurement
devices the error sources are essentially unknown. This, combined with the use of imperfect algorithms that are
not available to the end-user (i.e. "black-box" models) to minimise errors, means that, colocation data is often the
best option available to end-users in order to assess the applicability of a measurement method for their desired
purpose. This is particularly the case for LCS air pollution measurement devices. In this section, we show
colocation data collected as part of the UK Clean Air program funded QUANT project, and use the tools described
above to investigate the impact of the observed errors on end-use.
Figure 5 shows two colocated measurements from two different LCS devices: one measuring $NO_2$ (a-panels) and
the other $O_3$ (b-panels). Both measurements are compared with colocated reference measurements at an urban
background site in the city of Manchester. Unlike the modelled instruments in Sect. 4.1, the combination of error
sources is unknown in this case, and we can thus only assess the LCS measurement performance through
comparison with the reference measurements using global metrics and visual tools.
Single value metrics indicate an acceptable performance for both measurements: high linearity (both $R^2$ are higher
than 0.8) and relatively low errors (RMSE ~ 5ppb). However, the plots present the data in a variety of ways that
enable the user to identify patterns in the measurement errors that would be less obvious if only global metrics

were used. For example, the NO₂ sensor (LCS1, a-panels) has a non-linear response that is almost imperceptible from the regression plot but stands out in the B-A plot. Furthermore (despite the high $R^2$ and relatively low RMSE), the REU plot shows high relative errors that do not meet the Class 2 DQO for the measured concentration range. Regarding the O₃ sensor (LCS2, b-panels), the B-A plot shows two high density measurement clusters, one with positive absolute errors (over-measuring) and a larger one with negative errors (under-measuring). These are the result of a step change in the correction algorithm applied by the manufacturer and could easily have been missed if only summary metrics and a regression plot were used, especially if the density of the data points was not coloured.

It is worth noting that these plots do not directly identify the source of the proportional bias, with sensor response to the target compound or another covarying compound possible, but provides information on how much it impacts the data.

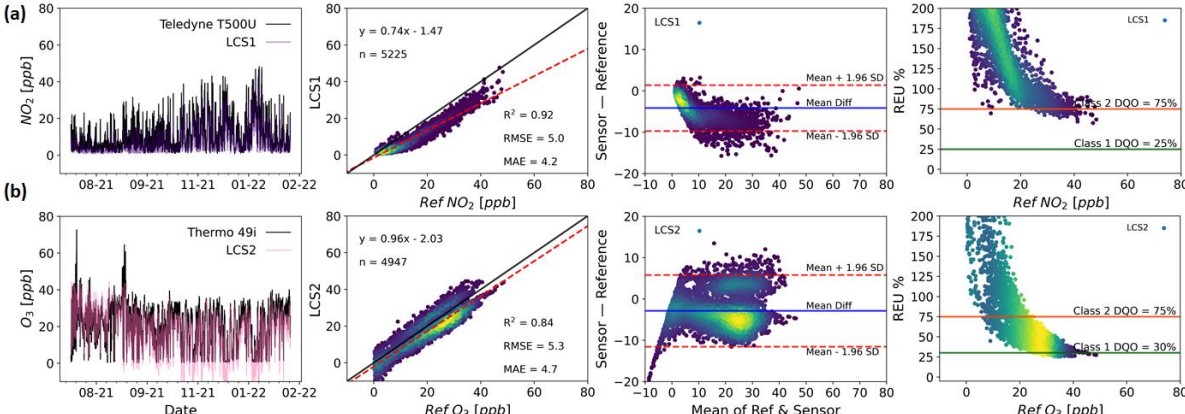

**Figure 5. Time series (left panels), regression plots (middle-left panels), Bland-Altman plots (middle-right panels) and REU (right panels; NO₂ Class 1 DQO = 25% & Class 2 DQO = 75% ; O₃ Class 1 DQO = 30% & Class 2 DQO = 75%) for NO₂ (a-panels) and O₃ (b-panels) measurements by two LCS systems of different brands in the same location and time span (Manchester Supersite, July 2021 to February 2022. Time res 1 h). All but the time-series plots, have coloured by data density.**

Figure 6 shows three out-of-the-box PM₂.₅ measurements made by two devices (LCS3 & LCS4) from the same brand in spring (LCS3: a-panels; LCS4: c-panels) and in autumn (b-panels, only LCS3). The colocation shown correspond to two different sites: an urban background site (LCS3, a and c-panels) and a roadside site (LCS4, c-panels).

As the regression and the B-A plots show, all LCS measurements in Fig. 6 have a proportional bias compared with the reference, with the LCS over predicting the reference values. The device at the urban background site (LCS3) show a dissimilar performance in spring and autumn, indicating that the errors this device suffers are differently influenced by local conditions in the two seasons (all the duplicates at the urban background show the same pattern). While for LCS3 during spring the error have a more linear behaviour, in autumn a non-linear pattern is clearly observed in the regression and B-A plots. Despite the utility that single metrics can have in certain

circumstances, the non-linear pattern goes completely unnoticed by them: while for the two different seasons RMSE and the MAE are almost constant the $R^2$ indicates a higher linearity for autumn.

A number of duplicates were deployed at both sites showing a very similar performance in terms of the single metric values but also in regard to the more visual tools (not shown here). This internal consistency is highly desirable, especially when LCS's are to be deployed in networks, as although mean absolute measurement error may be high, differences between identical devices are likely to be interpretable.

Having prior knowledge of the nature of the measurement errors allows informed experimental design prior to data collection. This is key if an end user is to maximise the power of a dataset, and the information it provides, to answer a specific question. For example, if an end-user wanted to identify pollution hotspots within a relatively small geographical area, then using a dense network of sensor devices that posses errors large and variable enough to make quantitative comparisons with limit values difficult (possibly due to an interference from a physical parameter like relative humidity) but show internal consistency could be a viable option. Providing the hotspot signal is large enough relative to any random error magnitude.

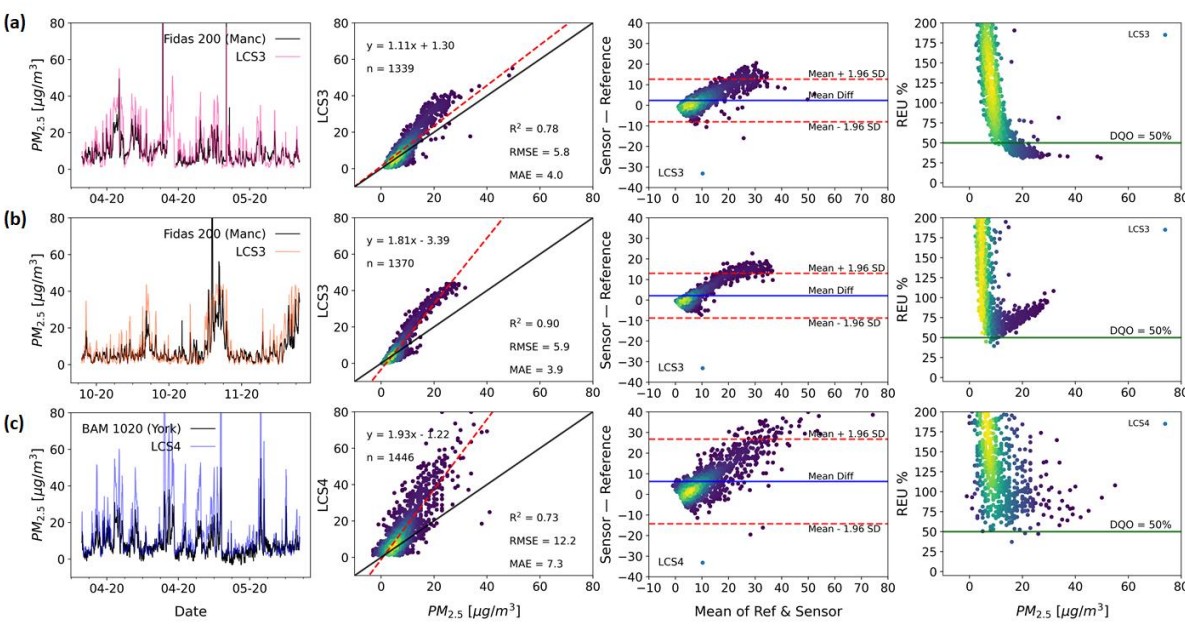

**Figure 6. Two LCS systems (LCS3 & LCS4, same brand) measuring PM2.5 (Time res 1 h). While LCS3 is shown for the same location (Manchester) but unfolded in two different seasons (a-panels: Apr to May 2020; b-panels: Oct to Nov 2020), LCS4 is at a different location (c-panels: York, Apr to May 2020). Time series (left panels), regression plots (middle-left panels), Bland-Altman plots (middle-right panels) and REU (right panels; DQO$_{PM2.5}$ = 50%) are used to characterise the device's error structure. All but the time-series plots have been coloured by data density.**

The LCS data from the roadside location (LCS4) show significantly lower precision than those at the urban background site, as seen in the B-A plot. This could be caused by differences in particle properties and size distributions between the two sites (Gramsch et al., 2021), and by the high frequency variation of transport

emissions close to the roadside site and turbulence effects (Baldauf et al., 2009; Makar et al., 2021). Duplicate
measurements show that all sensors of this type responded similarly in this roadside environment (not shown
here), supporting the high internal consistency of this device, but indicating a spatial heterogeneity in some key
error sources. It is also worth noting that the gold standard instruments at the two sites are not "reference method"
but "reference equivalent methods" (GDE, 2010), each using a different measurement technique: while an optical
spectrometer (Palas Fidas 200) is used in Manchester, the York instrument uses a Beta attenuation method (Met
One BAM 1020), which could also potentially lead to some of the observed differences. The increased apparent
random variability for LCS4, combined with the proportional bias, results in significantly higher measurement
uncertainty across the observed range, as can be seen by the REU plots, with LCS4 never reaching an acceptable
DQO level (50% for $PM_{2.5}$). If the observed proportional bias is corrected the linearly bias-corrected sensors (Fig.
S3) show a much improved comparison with the reference measurement, specially LCS3* in autumn and LCS4*.
The error distribution for the LCS3 (autumn) shown by the B-A plot is greatly narrowed (~3x times) and now the
sensor is accomplishing the DQO below 10 $ugm^{-3}$ as the REU plot indicates. For LCS4 the B-A plot shows an
error characteristic more dominated by random errors, and a significant reduction of the relative uncertainty, with
the REU at 10 $ugm^{-3}$ reducing from ~75 to ~50%.
As a comparison for the LCS data shown above, Fig. 7 shows two identical $NO_2$ reference grade instruments,
Teledyne T200U (Chemiluminescence method) at the Manchester urban background site (panels a and b) during
two different time periods, with a Teledyne T500U (CAPS detection method) used as the "ground truth"
instrument. Instrument "a" manifests a significant proportional bias, in contrast to instrument "b", but both show
differences that could be non-negligible depending on the application. The deviations observed in instrument "a"
was due to the cell pressure being above specification by ~20%, unnoticed while the instrument was in operation.
This demonstrates the importance of checking instrument parameters regularly in the field even if the data appears
reasonable.
As the LCS error structure is determined relative to the performance of a reference measurement, if the reference
instrument suffers from significant errors this will affect the outcomes of the performance assessment, due to the
assumption that all the errors reside with the LCS. As Fig. 7 shows, however, this assumption is not necessarily
always valid and potentially argues that reference instruments used in colocation studies should be subject to
further error characterisation, including possible colocation with other reference instruments. As a similar
comparison of reference instruments, Fig. S3 shows two ozone research grade instruments (a Thermo 49i and a
2B).
It is worth noting that even when using reference, or reference equivalent, grade instrumentation, inherent
measurement errors mean that relative uncertainty, as shown in the REU plot, increases asymptotically at lower
values. This is not unexpected, but is potentially important as ambient target concentration recommendations
continue to fall based on updated health evidence (World Health Organization, 2021).

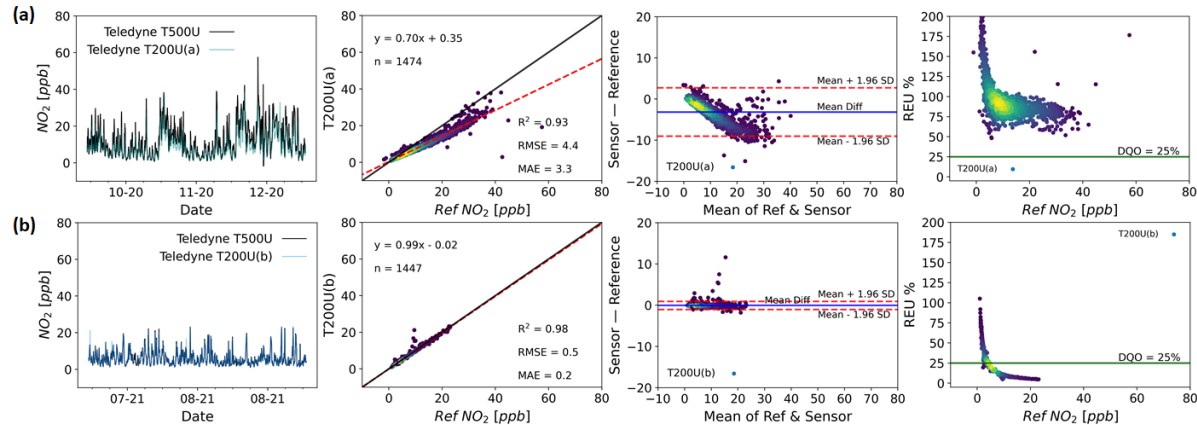

**Figure 7. Time series (left panel), regression plots (middle-left panel), Bland-Altman plots (middle-right panel) and REU (right panel, DQO for NO₂ = 25%) for two identical (Teledyne T200U) reference NO₂ instruments (panels a and b) colocated at the Manchester Supersite (1h time res). The first instrument between October & November 2020 and the second between July & August 2021. All but the time-series plots have been coloured by data density.**

## 5. Discussion

The widespread use of colocation studies to assess measurement device performance, means many examples exist in the LCS literature where different devices are compared using summary metrics for field or laboratory studies (Broday, 2017; Duvall et al., 2016; Hofman et al., 2022; Karagulian et al., 2019; Mueller et al., 2017; Rai et al., 2017; van Zoest et al., 2019). Although these comparisons do provide useful information, they can be misleading for end users wanting to compare the performance of different devices, as they are often carried out under different conditions and do not present the data or experimental design in full. Even in the case where comparisons have been done under identical conditions, the data still needs to be treated with caution, as inevitable differences between assessment environment and proposed application environment, as well as any changes to instrument/sensor design or data processing, mean that past performance does not guarantee future performance.

All measurement devices suffer from measurement errors, many of which are potentially significant depending on the application, with devices and their error susceptibility covering a broad spectrum. As evidenced by Fig. 7, reference instruments are not immune from this phenomena, with the proportional bias of one of the NOx instruments clearly affecting its measurements resulting in the absolute error increasing with concentration. As the requirements on measurement devices continue to increase, driven in part by new evidence supporting the reduction of air pollutant target values, the devices currently being used for a particular application could no longer be fit-for-purpose in the situation where the limit value has decreased to the point where it is small relative to the device's uncertainty.

Single value performance metrics, such as $R^2$ and RMSE, can seem convenient when comparing multiple co-located devices as they facilitate decision making when a threshold criterion is defined. However, these scalar values hide important information about the scale and / or distribution of the errors within a dataset; graphical summaries of the measurements themselves can offer significantly more insight into the impact of measurement errors on device performance and ultimate capabilities. Of particular use in air pollution measurements is the ability to see how the errors manifest themselves in relation to our best estimate of the true pollutant concentration, as often applications have specific target pollutant concentration ranges of interest. For example, the two LCS

devices shown in Fig. 5 have considerably high $R^2$ values (0.92 and 0.84) and relatively low RMSE and MAE,
but one suffering of non-linear errors (LCS1) and the other with data coming from two different calibration states
(LCS2).
Errors, or combinations of errors, frequently result in varying magnitude of the observed measurement
inaccuracies across the concentration space observed, and it is often useful to assess both the absolute and relative
effects of the errors. By getting a more complete picture of the device performance, decisions can be made on the
effectiveness of simple corrections, such as correcting for an apparent proportional bias using an assumption of a
linear error model. Ultimately end users need to identify the data requirements a priori and design quantifiable
success criteria by which to judge the data. For example, rather than just wanting to measure the 8-hour average
$NO_2$, be more specific and require that this needs to be accurate to within 5 ppb, have demonstrated approximately
normally distributed errors in a representative environment for the period of interest, and no statistical evidence
of deviation from a linear correlation with the reference measurement over the target concentration range for the
period of interest.
A major challenge comes from complex errors, such as interferences from other compounds or with environmental
factors, that vary temporally and/or spatially. Similar graphical techniques to those presented above can be used
to identify the existence of such relationships, but correcting for them remains a challenge. For example, the
correlation between measurement errors and relative humidity could be explored by replacing the abscissa with
measured relative humidity in both the B-A and REU plots. This would visualise the relationship between absolute
and relative errors with relative humidity, but would not be able to confirm causality. The complex and covarying
nature of the atmosphere means that the best way to identify a device error source is through controlled laboratory
experiments, where confounding variables can be controlled, although these experiments are often difficult and
expensive to perform in a relevant way.
This brings into question the power of colocation studies, as they can ultimately never be performed under the
exact conditions for every intended application. The $PM_{2.5}$ sensors shown in Fig. 6 demonstrate this, as if a
colocation dataset generated at the urban background site was used to inform a decision about the applicability of
these devices to a roadside monitoring task, then an overly optimistic assessment of the scale of the errors to be
expected would be likely. It is therefore always desirable that colocation studies are as relevant as possible to the
desired application, and this is even more paramount in the case where the error sources are poorly specified. For
this reason, complete meta-data on the range of conditions over which a study was conducted is key information
in judging its applicability to different users.
Although there is no strict definition on what makes a device a LCS, we often make the categorization based on
the hardware used. Standard reference measurement instruments are generally based on well-characterised
techniques developed and improved over years, based primarily on the progressive refinement of hardware (e.g.
materials used for the detection elements, electronic circuits to filter noise, refinement of production methods,
etc.). Although LCS sensor technologies are improving, it is interesting that many of the significant improvements
that have been made to LCS performance have been through software, rather than hardware advances. As more
colocation data are generated in different environments, many LCS manufacturers have been able to develop data
correction algorithms that minimise the scale of the errors that are present on the LCS hardware. This can greatly
improve the performance of LCS devices, and has been a large factor in the improvements seen in these devices
over recent years. These algorithms are, however, inevitably imperfect and can suffer from concept drift (De Vito
et al., 2020), caused by the lack of available colocation data over a full spectrum of atmospheric complexity.
Furthermore, any kind of statistical model introduces a new error source that can work in conjunction with the
pre-existing measurement errors to drastically change the observed error characteristics, making it much more
difficult for users to interpret and extrapolate from colocation study performance to intended application. If end
users are to be able to make well informed decisions about device applicability then information on the scale of
the measurement errors, and the impact of corrections made to minimise these, should be made available.
Exemplar case studies in a range of relevant environments would also be highly valuable. Unfortunately, this
colocation data are costly to generate, meaning relevant data often does not exist, and when it does is often not
communicated in such a way that enables the user to make a fully informed decision.
**6. Conclusions**
In situ measurements of air pollutants are central to our ability to identify and mitigate poor air quality.
Measurement applications are wide ranging, from assessing legal compliance to quantifying the impact of an
intervention. The range of available measurement tools for key pollutants is also increasingly broad, with
instrument price tags spreading several orders of magnitude. In order for a measurement device to be of use for a
particular application it must be fit-for-purpose, with cost, useability and data quality all needing to be considered.
Understanding measurement uncertainty is key in choosing the correct tool for the job, but in order for this to be
assessed the job needs to be fully specified a priori. The specific data requirements of each measurement
application need to be understood and a measurement solution chosen that is capable of providing data with
sufficient information content.
In order to aid end users in extrapolating from colocation study performance to potential performance in a specific
application, performance metrics are often used. Although single value performance metrics do convey some
useful information about the agreement between the data from the measurement device being assessed and the
reference data, they can often be misleading in their evaluation of performance. This dictates a more rigorous and
empirical approach to data uncertainty assessment in order to determine if a measurement is fit for purpose. The
ability to assess device performance across the observed concentration range, as in the B-A and REU plots, enables
an end-user to make an informed decision about the capabilities of a measurement device in the target
concentration range. These visual tools also help identify any simple corrections that can be applied to improve
performance. In contrast, if an end-user was only provided with a single value metric, such as $R^2$ or RMSE then
it would be significantly more difficult to understand the likely implications of the measurement uncertainties.
All measurement devices suffer from errors, which result in deviations between the reported and true values.
These errors can come from a multitude of sources, with the scale of the deviation from the true value being
dependent on the nature of the error. Although a known measurement uncertainty for all applications would be
ideal for end users to be able to assess measurement device suitability for purpose, in many cases, especially for
LCS, this is not possible due to the presence of poorly characterised, or sometimes unknown, error sources. In the
absence of this, useful information on likely measurement performance can be obtained using colocation data
compared with a measurement with a quantified uncertainty. It is important that such a colocation study is carried

out in an environment as similar as possible to the application environment, as the unknown nature of many error sources means their magnitude can change significantly between different locations and/or seasons (e.g. Fig. 6). Ideally, depending on the measurement task, the user could use the colocation data to model the error causes and use this to develop strategies to minimise final measurement uncertainty. Unfortunately, relevant colocation study are often not available, and to generate the data would be prohibitively costly, which limits the user's ability to make a realistic assessment of likely uncertainties. The presence of, often complex, error minimisation post processing or calibration algorithms further complicates things. This additional uncertainty is most likely to bias any performance prediction if the end user is unaware of the purpose or scale of the data corrections, and their applicability to the target environmental conditions. Ideally, long term colocation data sets demonstrating the performance of measurement hardware and software, in a range of relevant locations, over multiple seasons, and carried out by impartial bodies would be available to inform measurement solution decisions.

In order for end users to take full advantage of the ever increasing range of air pollution measurement devices available, the questions being asked of the data must be consummate with the information content of the data. Ultimately this information content is determined by the measurement uncertainty. Thus, providing end users with as accurate an estimate as possible of the likely measurement uncertainty, in any specific application, is essential if end users are to be able to make informed decisions. Similarly, end users must specify the data uncertainty requirements for each specific task if the correct tool for the job is to be identified. This requirement for air quality management strategies to acknowledge the capabilities of available devices, both in the setting and monitoring of limits, will only become increasingly important as target levels continue to decrease.

**Supplementary**

The supplement related to this article is available online at:

**Code and data availability**

The code and data for this study can be found on Zenodo: https://zenodo.org/record/6518027#.YnKbH9PMJhE. The live code can be found on GitHub: https://github.com/wacl-york/quant-air-pollution-measurement-errors.

**Author contributions**

PE: Funding acquisition; Supervision. SD and PE: Project administration; Formal analysis. SD, PE & SL: Conceptualization; Methodology; Investigation. SD & SL: Visualisation; Software. KR, NM, MF: Resources. SD, SL, KR, NM, MF: Data curation. SD, PE, SL, TB, NM, TG & DH: Writing – review & editing.

**Competing interests**

The authors declare that they have no conflict of interest.

**Acknowledgements**

This work was funded as part of the UKRI Strategic Priorities Fund Clean Air program (NERC NE/T00195X/1), with support from Defra. We would also thank the OSCA team (Integrated Research Observation System for Clean Air) at the Manchester Air Quality Supersite (MAQS), for help in data collection for the regulatory-grade

instruments. The secondary research grade instruments used here (Thermo ozone 49i, 2B Technologies 202 ozone, and Teledyne T200U NOx) are provided through the Atmospheric Measurement and Observation Facility (AMOF) and the calibrations were carried out in the COZI Laboratory, a facility housed at the Wolfson Atmospheric Chemistry Laboratories (WACL). Both funded through the National Centre for Atmospheric Science (NCAS). Special thanks to Elena Martin Arenos, Chris Anthony, Killian Murphy, Stuart Young, Steve Andrews and Jenny Hudson-Bell from WACL for the help and support to the project. Also thanks to Stuart Murray and Chris Rhodes from the Department of Chemistry Workshop for their technical assistance and advice. Thanks to Andrew Gillah and Michael Golightly from the York Council who assisted with site access.

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
